# Learning Semantic Role Labeling from Compatible Label Sequences

**Tao Li**[*]
Google Research
tlinlp@google.com

**Ghazaleh Kazeminejad**
University of Colorado Boulder
ghka9436@colorado.edu

**Susan W. Brown**
University of Colorado Boulder
susan.brown@colorado.edu

**Martha Palmer**
University of Colorado Boulder
martha.palmer@colorado.edu

**Vivek Srikumar**
University of Utah
svivek@cs.utah.edu

## Abstract

Semantic role labeling (SRL) has multiple disjoint label sets, e.g., VerbNet and PropBank. Creating these datasets is challenging, therefore a natural question is how to use each one to help the other. Prior work has shown that cross-task interaction helps, but only explored multitask learning so far. A common issue with multitask setup is that argument sequences are still separately decoded, running the risk of generating structurally *inconsistent* label sequences (as per lexicons like SEMLINK). In this paper, we eliminate such issue with a framework that jointly models VerbNet and PropBank labels as one sequence. In this setup, we show that enforcing SEMLINK constraints during decoding constantly improves the overall F1. With special input constructions, our joint model infers VerbNet arguments from given PropBank arguments with over 99 F1. For learning, we propose a constrained marginal model that learns with knowledge defined in SEMLINK to further benefit from the large amounts of PropBank-only data. On the joint benchmark based on CoNLL05, our models achieve state-of-the-art F1's, outperforming the prior best in-domain model by 3.5 (VerbNet) and 0.8 (PropBank). For out-of-domain generalization, our models surpass the prior best by 3.4 (VerbNet) and 0.2 (PropBank).

## 1 Introduction

Semantic Role Labeling (SRL, Palmer et al., 2010) aims to understand the role of words or phrases in a sentence. It has facilitated other natural language processing tasks including question answering (FitzGerald et al., 2018), sentiment analysis (Marasović and Frank, 2018), information extraction (Solawetz and Larson, 2021), and machine translation (Rapp, 2022).

Semantic role labeling can take various forms, each associated with different datasets. Predicates can be coarsely divided into PropBank (Palmer et al., 2005) senses, each with a core set of numbered semantic arguments (e.g., ARG0–ARG5). There are also modifier arguments (e.g., ARGM-LOC) typically representing information such as the location, purpose, manner or time of an event. Alternatively, predicates can also be hierarchically clustered into VerbNet (Schuler, 2005) classes according to similarities in their syntactic behavior. Each class admits a set of thematic roles (e.g. AGENT, THEME) whose interpretations are consistent with all predicates within the class.

As a modeling problem, SRL requires associating argument types and phrases with respect to an identified predicate. The two labeling tasks (i.e., VerbNet SRL and PropBank SRL) are closely related; but they differ in their treatment of predicates and have disjoint label sets. Learning jointly can improve data efficiency across the different labeling SRL tasks.

A common formulation used to instantiate this idea in prior work is multitask learning (e.g. Strzyz et al., 2019; Gung and Palmer, 2021): each label set is treated as a separate labeling task, and sometimes also modeled with inter-task feature interaction or consistency losses. While multitask learning often works well in such cases, the loss formulation represents a conservative view over label compatibilities of different tasks. At prediction time, subtask modules still run independently and are not constrained by each other. Consequently, decoded labels may violate structural constraints with respect to each other. In such settings, constrained inference (e.g., Fürstenau and Lapata, 2012; Greenberg et al., 2018) has been found helpful. However, this raises the question of how to involve such inference during learning for better data efficiency. Furthermore, given the wider availability of PropBank-only data (e.g., Pradhan et al., 2013, 2022), how to efficiently benefit from such data also remains a question.

---

*Work done at the University of Utah.

In this paper, we argue that the two disjoint but compatible labeling tasks can be more effectively modeled as one task using their compatibility structures that are already explicitly defined in the form of SEMLINK (Stowe et al., 2021). SEMLINK offers mappings between various semantic ontologies including PropBank and VerbNet. Gung and Palmer (2021) devised a deterministic conversion from PropBank label sequences to VerbNet ones using only the unambiguous mappings in SEMLINK. This conversion gives a test bed that has half of the predicates in CoNLL05 SRL dataset (Carreras and Màrquez, 2005) with both VerbNet and PropBank jointly labeled.

Given this setting, we propose a simple and effective joint CRF model for the VerbNet SRL and PropBank SRL tasks. In addition to the joint CRF, we propose an inference constraint that uses compatible label structures defined in SEMLINK, and show that our constrained inference achieves higher overall SRL F1—the average of VerbNet and ProbBank F1 scores—than the current state-of-the-art. Indeed, when PropBank labels are observed, it achieves over 99 F1 on the VerbNet SRL, suggesting the possibility of an automated annotation helper. We show that our formulation naturally extends to a constrained marginal model that learns from the more abundant PropBank-only data in a semi-supervised setting. When learning and predicting with constraints, it achieves even better SRL F1 in out-of-domain generalization.[1]

## 2  Joint Task of Semantic Role Labeling

We consider modeling VerbNet (VN) and PropBank (PB) SRL as a joint labeling task. Given a sentence $x$, we want to identify a set of predicates (e.g., verbs), and for each predicate, generate two sequences of labels, one for VerbNet arguments $y^V$, the other for PropBank arguments $y^P$. With respect to VN parsing, a predicate is associated with a VerbNet class that represents a group of verbs with shared semantic and syntactic behavior, thereby scoping a set of thematic roles for the class. Similarly, the predicate is associated with a PropBank sense tag that defines a set of PB core arguments along with their modifiers. A example is shown in Tab. 1.

We treat predicate classification and argument

---

[1]The code for reproducing our experiments: https://github.com/utahnlp/marginal_srl_with_semlink

| $\eta$ | $\sigma$ | Admissible alignments ($y^V$-$y^P$) |
|---|---|---|
| price-54.4 | 01 | Agent-Arg0; Theme-Arg1; Value-Arg2 |
| price-54.4 | 02 | Agent-Arg0; Theme-Arg1; Value-Arg2 |
| admire-31.2 | 02 | Experiencer-Arg0; Stimulus-Arg1 |

Table 1: VN-PB alignments defined in SEMLINK for predicate *value*. $\eta$: VN class. $\sigma$: PB sense.

labeling as separate tasks and focus on the latter.[2] Assuming predicates $u$ and their associated VN classes $\eta$ and PB senses $\sigma$ are given along with $x$, we can write the prediction problem as:

$$(x, u, \eta, \sigma) \rightarrow (y^V, y^P) \qquad (1)$$

### 2.1  VerbNet Completion

There is a much larger amount of PropBank-only data (e.g., Pradhan et al., 2013, 2022) than jointly labeled data. Inferring VerbNet labels from observed PropBank labels, therefore, is a realistic use case. This corresponds to the modeling problem:

$$(x, u, \eta, \sigma, y^P) \rightarrow y^V \qquad (2)$$

We refer to this scenario as *completion* mode. In this paper, we will focus on the joint task defined in Eq. 1 while also generalizing our approach to address the completion task in Eq. 2.

### 2.2  Multitask learning and Its Limitations

When predicting multiple label sequences for SRL, a common approach is multitask learning using dedicated classifiers for each task that operate on a shared representation. The current state-of-the-art model (Gung and Palmer, 2021) used an LSTM stacked on top of BERT (Devlin et al., 2019) to model both PropBank and VerbNet. While each set of the semantic roles is modeled jointly with VerbNet predicates, the argument labeling of the two subtasks is still kept separate.

Separate modeling of VerbNet SRL and PropBank SRL has a clear disadvantage: subtask argument labels might disagree in three ways: 1) in terms of the BIO tagging scheme—e.g., a word having a B-* VN label and a I-* PropBank label, or 2) assigning semantically invalid label pairs—e.g., an ARGM-LOC being called a THEME, or 3) violating SEMLINK constraints. In Sec. 6, we show that a model with separate task classifiers, while having a close to state-of-the-art F1, can have a fair

---

[2]Prior work (e.g., Täckström et al., 2015) has shown that the predicate disambiguation can be modeled with high accuracy using standalone classifiers.

amount of argument assignment errors with respect to SEMLINK, especially for out-of-domain inputs.

## 3 A Joint CRF Model

To eliminate the errors discussed in Sec. 2.2, we propose to model the disjoint SRL tasks using a joint set of labels. This involves converting multi-task modeling into a single sequence labeling task whose labels are pairs of PB and VN labels. Doing so not only eliminates the BIO inconsistency, but also exposes an interface for injecting SEMLINK constraints.

Our model uses ROBERTA (Liu et al., 2019) as the backbone to handle textual encoding, similarly to the SRL model of Li et al. (2020). At a high level, we use a stack of linear layers with GELU activations (Hendrycks and Gimpel, 2016) to encode tokens to be classified for a predicate. For the problem of predicting arguments of a predicate $u$, we have an encoding vector $\phi_{u,i}$ for the $i$-th word in the input text $x$.

$$e = \mathsf{map}(\mathrm{ROBERTA}(x)) \quad (3)$$
$$\phi_u = \{f_{ua}\left([f_u(e_u), f_a(e_i)]\right), \forall_i \in x\} \quad (4)$$

Here, map sums up word-piece embeddings to form a sequence of word-level embeddings, the functions $f_u$ and $f_a$ are both linear layers, and $f_{ua}$ denotes a two-layer network with GELU activations in the hidden layer. We use a dedicated module of the form in Eq. 4 for the VN and PB subtasks. This gives us a sequence of vectors $\phi_u^v$ for VN and a sequence of vectors $\phi_u^p$ for PB.

Next, we project the VN and PB feature sequences into a $|Y^V| \times |Y^P|$ label space:

$$z_u = \{g([\phi_{u,i}^V, \phi_{u,i}^P]), \forall_i \in x\} \quad (5)$$

Here, $g$ is another two-layer GELU network followed by a linear projection that outputs $|Y^V| \times |Y^P|$ scores, corresponding to VN-PB label pairs. The final result $z_u$ denotes a sequence of VN-PB label scores for a specific predicate $u$. In addition, we use a CRF as a standard first-order sequence model over $z_u$ (treating it as emission scores), and use Viterbi decoding for inference. The training objective is to maximize:

$$\log P(y^{VP}|x) = s(y^{VP}, x) - \log Z(x) \quad (6)$$

where $s(\cdot)$ denotes the scoring function for a label sequence that adds up the emission and the transition scores, and the term $Z(x)$ denotes the partition that sums exponentiated scores over all label sequences. The term $y^{VP}$ denotes the label sequence that has VN and PB jointly labeled. We will refer to this model as the **joint CRF**, and the label sequence as the **joint labels**.

**Reduced Joint Label Space.** We use the cross-product the two label sets, prefixed with a BIO prefix[3]. A brute-force cross product leads to a $|Y^V| \times |Y^P|$ label space. In practice, it is important to keep the joint label space at a small scale for efficient computation, especially for the CRF module. Therefore, we condense it by first disallowing pairs of the form (B-*, I-*) and predicate-to-argument pairs. The former enforces that the VerbNet arguments do not start within ProbBank arguments, while the latter ensures that the predicate is not part of any argument. Next, we observe the co-occurence pattern of VN and PB arguments, disabling semantically invalid pairs such as (THEME, ARGM-LOC)[4]. This reduces the label space by an order of magnitude (from $144 \times 105 = 15,120$ to $685$).

**Input Construction using Predicates.** We take inspiration from prior work (e.g. Zhou and Xu, 2015; He et al., 2017; Zhang et al., 2022) to explicitly put predicate features as part of the input to augment textual information. At the same time, we also seek to maintain a simple construction that can be easily adapted to a semi-supervised setting (i.e. compatible with PropBank-only data). To this end, we propose a simple solution that appends the PropBank senses of potential predicates to the original sentence $x$:

$$x_{\mathrm{WP}} = [\mathrm{CLS}\ w_{1:T}\ \mathrm{SEP}\ \sigma_{1:N}\ \mathrm{SEP}]$$

where $w_{1:T}$ denotes the input words, and $\sigma_{1:N}$ denotes the senses of the $N$ predicates. In practice, we use the PropBank roleset IDs which consist of a pair of (lemma, sense)—e.g., *run.01*. Our models only take the encodings for $w_{1:T}$ after the ROBERTA encoder and ignore the rest. We consider this design to be more efficient than prior work (e.g. Gung and Palmer, 2021; Zhang et al., 2022) that dedicated text feature for each predicate. In our setup, the argument labeling for different predicates shares the same input, thus no

---

[3]For example, the pair (B-THEME, B-ARG1).

[4]VerbNet arguments typically align to PropBank core arguments, not modifiers; thus pairs like (ACTOR, ARGM-CAU) can be filtered out.

need to run encoding multiple times for multiple predicates.

## 4 Semi-supervised Learning with PropBank-only Data

Compared to data with both VerbNet and Prop-Bank fully annotated, there is more data with only PropBank labeled. The SEMLINK corpus helps in unambiguously mapping $\sim 56\%$ of the CoNLL05 data (Gung and Palmer, 2021). Therefore, a natural question is: *can we use PropBank-only data to improve the joint task?*

Here, we explore model variants based on the joint CRF architecture described in Sec. 3. We will focus on modeling for PropBank-only sentences.

### 4.1 Separate Classifiers for VN and PB

As a first baseline, we treat VN and PB as two separate label sequences during training. This is essentially a multitask setup where VN and PB targets use separate classifiers. We let these two classifiers share the same ROBERTA encoder, and have their own learnable weights for Eq. 4-6.

### 4.2 Dedicated PropBank Classifier

Another option is to retain the joint CRF for the jointly labeled dataset and use an additional dedicated CRF for PB-only sentences. Note that this setup is different from the model in Sec. 4.1. As before, we let these two to share the same encoder and, for Eq. 4-6, they have dedicated trainable weights.

During inference, we rely on the Viterbi decoding associated with the joint CRF module to make predictions. In our preliminary experiments, the joint CRF and the dedicated PropBank CRF achieve similar F1 on PropBank arguments.

### 4.3 Marginal CRF

For partially labeled sequences, we take inspiration from Greenberg et al. (2018) to maximize the marginal distribution of those observed labels. In our joint CRF, the marginalization assumes uniform distribution over VN arguments that are paired with observed PB arguments. The learning objective is to maximize the probabilities of such label sequences as a whole:

$$\text{LSE}_{y \in y_u^P}\left(s(y)\right) - \log Z(x) \qquad (7)$$
$$\text{where } \text{LSE}_y\left(\cdot\right) = \log \sum_y \exp(\cdot)$$

where $y \in y_u^P$ denotes a potential joint label sequence with only PropBank arguments observed for a predicate $u$. Scores of such label sequences are aggregated by the LSE operator. Note that the marginal CRF and the joint CRF (Eq. 6) use the same model architecture, just with a different loss.

### 4.4 Marginal Model with SEMLINK (Marginal_SEML)

The log marginal probability in Eq. 7 assumes uniform distribution over a large label space. It included any arbitrary VerbNet arguments paired to the observed PropBank labels. In practice, we can narrow it down to only legitimate VN-PB argument pairs defined in SEMLINK. Such legitimate space is uniquely determined by a VerbNet class $\eta$ and PropBank sense $\sigma$. We will refer to label sequences that comply with this space as $y_u^{\text{SEML}}$, and apply it on Eq. 7:

$$\text{LSE}_{y \in y_u^P \cap y_u^{\text{SEML}}} s(y) - \text{LSE}_{y \in y_u^{\text{SEML}}} s(y) \qquad (8)$$

Note that this formulation essentially changes the global optimization into a local version which implicitly requires using SEMLINK at inference time. We will present the details of $y_u^{\text{SEML}}$ in Sec. 5. Intuitively, it zeros out losses associated with joint label traces that violate SEMLINK constraints. During training, we found that it is important to apply this constraint to both B-* and I-* labels.

**Where to apply $y_u^{\text{SEML}}$?** Technically, the summation over reduced label space can be applied at different places, such as the partition $Z$ in Eq. 6. We will report performances on this setting in Sec. 7.2. In short, plugging the label filter $y_u^{\text{SEML}}$ to the joint CRF (therefore jointly labeled data) has little impact on F1 scores, thus we reserve it for the PropBank-only data (as in Eq. 8).

## 5 Inference with SEMLINK

Here we discuss the implementation of the $y_u^{\text{SEML}}$. Remember that each pair of VerbNet class $\eta$ and PropBank sense $\sigma$ uniquely determines a set of joint argument labels for the predicate $u$. For brevity, let us denote this set as SEML($u$) (e.g., Tab. 1). Eventually, we want the Viterbi-decoded

label sequence to comply with $\text{SEML}(u)$. That is,

$$\forall(l^V, l^P) \in \text{SEML}(u) \rightarrow$$
$$\forall i, \Big[ \big( \forall_{l \notin \text{SEML}(u)}(y_i^V = l^V, l) \notin y_u^{VP} \big)$$
$$\wedge \big( \forall_{l \notin \text{SEML}(u)}(l, y_i^P = l^P) \notin y_u^{VP} \big) \Big]$$
$$(9)$$

where $i$ denotes the location in a sentence, $y_u^{VP}$ is the joint label sequence, consisting of $(y_i^V, y_i^P)$ pairs. The constraint in Eq. 9 translates as: *if a VerbNet argument is present in the predicate $u$'s* SEMLINK *entry, we prevent it from aligning to any PropBank arguments not defined in* SEMLINK; *and the same applies to PropBank arguments.*

This constraint can be easily implemented by a masking operation on the emission scores of the joint CRF, thus can be used *at both training and inference time*. During inference, it effectively ignores those label sequences with SEMLINK violations during Viterbi decoding:

$$y_u^{VP} = \arg\max_{y \in y_u^{\text{SEML}}} s(y) \qquad (10)$$

In Sec. 6, we will show that using Eq. 10 always improves the overall SRL F1 scores.

**VerbNet Label Completion.** For models based on our joint CRF, we mask out joint labels that are not defined in $y^P$ during inference, similar to Eq. 8. For models with separate VN and PB classifiers (in Sec. 4.1), we enforce the VN's Viterbi decoding to only search arguments that are compatible with the gold $y^P$ in SEMLINK. Furthermore, we always use the constraint (Eq. 9) in the completion mode.

## 6 Experiments

In this section, we aim to verify whether the compatibility structure between VerbNet and PropBank (in the form of SEMLINK) has a positive impact on their sequence labeling performances.

### 6.1 Data

We follow the prior state-of-the-art (Gung and Palmer, 2021) in extracting VerbNet labels from the CoNLL05 dataset using the SEMLINK corpus. We use the same version of SEMLINK to extract the data for training and evaluation. Therefore, our F1 scores are directly comparable with theirs (denoted as IWCS2021 in Table 2) The resulting dataset accounts for about $56\%$ of the CoNLL05 predicates, across training, development and test

sets (including WSJ and Brown). We will refer to this data as the joint data column in Table 2. For semi-supervised learning, we incorporate the rest of the PropBank-only predicates in the CoNLL05 training split. For development and testing, we use the splits in the joint dataset for fair comparison with prior work.

### 6.2 Training and Evaluation

We adopt the same fine-tuning strategy as in Li et al. (2020)—we fine-tune twice since this generally outperforms fine-tuning only once, even with the same number of total epochs. In the first round, we fine-tune our model for 20 epochs. In the second round, we restart the optimizer and learning rate scheduler, and fine-tune for 5 epochs. In both rounds, checkpoints with the highest average VN/PB development F1 are saved. For different model variants, we report the average F1 from models trained with 3 random seeds.

For the SEMLINK constraint, we use the official mapping between VN/PB arguments[5]. When involved in constrained inference, we use the gold VerbNet classes $\eta$ and PropBank senses $\sigma$.

For evaluation, in addition to the standard VN and PB F1 scores, we also report the percent of predicates with predictions that are inconsistent with SEMLINK, denoted by $\rho$.

### 6.3 Performance on SEMLINK Extracted CoNLL05

We want to compare models trained on the joint dataset and variants (in Sec. 4) on the semi-supervised setup. Table 2 presents their performances along with SEMLINK violation rates in model predictions. Note that the ground truth joint data bears no violation at all (i.e., $\rho = 0$).

**Multitask involves SEMLINK violations.** Firstly, we show the limitations of multitask learning. While the architecture is simple, the testing scores mostly outperform the previous state-of-the-art, except the Brown PropBank F1. However, there is a fair percentage of predicates having structurally wrong predictions, especially in the Brown test set. With semi-supervised learning, VN and PB F1s are improved on the Brown set while slightly lowered on the WSJ VN. This also comes with a degraded SEMLINK error rate ($3.43 \rightarrow 4.08$ on WSJ and $8.71 \rightarrow 10.48$

---

[5]https://github.com/cu-clear/semlink/blob/master/instances/semlink-2

| Data | Train Fine-tune×2 | Inf SEML | Dev | | | WSJ | | | Brown | | |
|---|---|---|---|---|---|---|---|---|---|---|---|
| | | | VN | PB | $\rho\downarrow$ | VN | PB | $\rho\downarrow$ | VN | PB | $\rho\downarrow$ |
| Joint | IWCS2021 (gold $\eta$ & $\sigma$) | - | - | - | - | 88.2 | 88.7 | - | 83.0 | 82.8 | - |
| | Multitask | ✗ | $89.51_{.09}$ | $87.91_{.17}$ | $3.68_{.06}$ | $90.74_{.23}$ | $89.33_{.15}$ | $3.43_{.50}$ | $83.69_{1.54}$ | $81.73_{.98}$ | $8.71_{.74}$ |
| | Joint | ✓ | $90.20_{.22}$ | $87.58_{.18}$ | 0 | $91.42_{.11}$ | $89.12_{.16}$ | 0 | $85.90_{1.11}$ | $82.56_{.52}$ | 0 |
| Semi | Multitask | ✗ | $89.56_{.09}$ | $88.30_{.42}$ | $4.82_{.35}$ | $90.45_{.09}$ | $89.34_{.74}$ | $4.08_{.37}$ | $84.28_{.85}$ | $82.51_{.70}$ | $10.48_{.16}$ |
| | Joint+CRF$_{\text{PB}}$ | ✓ | $90.73_{.32}$ | $88.42_{.11}$ | 0 | $91.52_{.36}$ | $89.30_{.29}$ | 0 | $85.72_{.13}$ | $82.33_{.31}$ | 0 |
| | Marginal | ✓ | $\mathbf{90.91}_{.27}$ | $\mathbf{88.68}_{.11}$ | 0 | $\mathbf{91.74}_{.22}$ | $\mathbf{89.51}_{.28}$ | 0 | $85.39_{1.74}$ | $82.46_{1.06}$ | 0 |
| | Marginal$_{\text{SEML}}$ | ✓ | $90.87_{.32}$ | $88.59_{.24}$ | 0 | $91.55_{.28}$ | $89.49_{.18}$ | 0 | $\mathbf{86.39}_{.17}$ | $\mathbf{83.04}_{.37}$ | 0 |

Table 2: Model performances on SEMLINK extracted CoNLL05 data. Each data point represents the mean$_{\text{std}}$ of 3 random runs. $\rho$: percentage of predicates with argument prediction that violates SEMLINK structures. IWCS2021 numbers are reported in (Gung and Palmer, 2021).

| Data | Train Fine-tune×2 | Inf SEML | Dev | | | WSJ | | | Brown | | |
|---|---|---|---|---|---|---|---|---|---|---|---|
| | | | VN | PB | $\rho\downarrow$ | VN | PB | $\rho\downarrow$ | VN | PB | $\rho\downarrow$ |
| Joint | Joint | ✗ | 89.28 | **87.70** | 1.58 | 90.73 | 89.10 | 1.59 | 84.43 | 82.48 | 5.99 |
| | Joint | ✓ | **90.20** | 87.58 | 0 | **91.42** | **89.12** | 0 | **85.90** | **82.56** | 0 |
| Semi | Joint+CRF$_{\text{PB}}$ | ✗ | 89.66 | 88.31 | 2.74 | 90.61 | 89.21 | 2.37 | 83.53 | 81.87 | 7.59 |
| | Joint+CRF$_{\text{PB}}$ | ✓ | **90.73** | **88.42** | 0 | **91.52** | **89.30** | 0 | **85.72** | **82.33** | 0 |
| | Marginal | ✗ | 89.79 | 88.50 | 2.92 | 90.73 | 89.30 | 2.39 | 83.23 | 81.88 | 6.74 |
| | Marginal | ✓ | **90.91** | **88.68** | 0 | **91.74** | **89.51** | 0 | **85.39** | **82.46** | 0 |
| | Marginal$_{\text{SEML}}$ | ✗ | 89.53 | 88.55 | 2.29 | 90.70 | 89.46 | 1.98 | 83.64 | 82.71 | 7.77 |
| | Marginal$_{\text{SEML}}$ | ✓ | **90.87** | **88.59** | 0 | **91.55** | **89.49** | 0 | **86.39** | **83.04** | 0 |

Table 3: Impact of SEMLINK at inference time. Each data point represents the average of 3 random runs. Improvements on VN F1s are both substantial and significant (p-value $\ll 0.01$).

on Brown). While SEMLINK inconsistency is not reported in (Gung and Palmer, 2021), we believe that, due to the nature of multitask learning, SEMLINK errors are inevitable.

**Joint CRF outperforms multitask learning.** A direct comparison is between the *Multitask* v.s. *Joint*. Our joint CRF obtains higher overall SRL F1 across the WSJ and the Brown sets. A similar observation applies to the semi-supervised setting where *Multitask* compares to *Joint+CRF$_{PB}$*. Most of such improvements are from the use of inference-time SEMLINK constraints.

**Inference with SEMLINK improves SRL.** In Table 3, we do side-to-side comparison of using versus not using the SEMLINK structure during inference. We do so for each modeling variant. With constrained inference, models no longer have SEMLINK structural violations ($\rho = 0$). And this results in a clear trend where using SEMLINK systematically improves the F1 scores. We hypothesize this is due to the reduced search space which makes

the decoding easier. Likely due to the higher granularity of VerbNet argument types compared to PropBank, a majority of the improvements are on the VN F1s.

**Does semi-supervised learning make a difference?** The answer is that it depends. For *Multitask*, using PropBank-only data traded off a bit on the overall WSJ F1 but improved the out-of-domain performances. Accompanied with this trade-off is the slightly higher inconsistency rate $\rho$. The *Joint+CRF$_{PB}$* model tells an opposite story that the partially labeled data is favorable on the in-domain test but not so in the out-of-domain test. This observation is also consistent with both the *Marginal* CRF and constrained Marginal model (*Marginal$_{SEML}$*). Furthermore, when performance improves, the margins on VN and PB are fairly distributed. Finally, we should note that, neither the *Joint+CRF$_{PB}$* nor the *Marginal* have a better Brown F1 than the *Joint*, meaning that they did not use the PB-only data efficiently.

**Impact of marginal CRF.** We compare the *Joint+CRF*$_{\text{PB}}$ to *Marginal* to see how a single CRF handles partially labeled data. The latter outperforms the former consistently by 0.2 on the in-domain test set but performed slightly worse on the out-of-domain Brown set. Comparing to the *Joint* model, it seems that naively applying marginal CRF leads to even worse generalization.

**Constrained marginal model improves generalization.** We want to see if our constrained model can help learning. Modeling PropBank-only data with a separate classifier (i.e., the *Multitask* and *Joint+CRF*$_{\text{PB}}$) failed to do so (although they indeed work better on the in-domain WSJ). In contrast, our constrained *Marginal*$_{\text{SEML}}$ apparently learns from the PropBank-only data more efficiently, achieving strong in-domain performance and substantially better out-of-domain generalization. This suggests that learning with constraints works better with partially labeled data. Interestingly though, it seems that the constraint is optional for fully annotated data since the *Marginal*$_{\text{SEML}}$ only enables the constraint on PB-only data. We verify this phenomenon in Sec. 7.2 with an ablation study.

**Statistical Significance of Constrained Inference.** We measure statistical significance using a t-test implemented by Dror et al. (2018) on predictions from models in Table 3. For each model, we compare inference with and without SEMLINK constraints (Sec. 5). For a fair comparison, we limit predictions from the model trained with the same random seed in each test, and apply the test for all random seeds (3 in total). We observe that the improvements on VerbNet F1's are universally significant. The p-values are far less than 0.01 across different testing data, models, and random seeds. This aligns to the observation in Table 3 that VN F1 has a substantial F1 boost while PB F1 improvements tend to be marginal.

To look closer, we examined the predictions of a *Joint* model on the Dev set (1,794 predicates), and found that, after using SEMLINK during inference, 51 wrongly predicted predicates in VN SRL were corrected (i.e., improved predicate-wise F1), and no predicates received a degraded F1. However, for PropBank SRL, there were 12 predicates corrected by the constraint while 6 became errors.

## 6.4 VerbNet Label Completion from PropBank

As discussed in Sec. 1, we also aim to address the realistic use case of VerbNet completion. Table 4 summarizes the performances for VerbNet argument prediction when gold PropBank arguments are given. In the completion mode, the *Joint* model performs generally better than all the semi-supervised models. This phenomenon is likely because the *Joint* model is optimized for the probability $P(y^V, y^P \mid x)$, while the semi-supervised models, in one way or the other, have a term for $\sum_{y^V} P(y^V, y^P \mid x)$ on PB-only data. The latter term does not explicitly boost model's discriminative capability on the unique ground truth.

In addition to the $x_{\text{WP}}$ input construction in Sec. 3, we propose a special construction $x_{\text{COMP}}$ for the VN completion mode by using the PB arguments as text input.:

$$x_{\text{COMP}} = [\text{CLS } w_{1:T} \text{ SEP } y_1^P ... \sigma_v \text{ } y_{v+1}^P ... \text{ SEP}]$$

where $y_i^P$ denotes the PropBank argument label for the $i$-th word. For the predicate word, we use the predicate feature (i.e. lemma and sense). Compared to $x_{\text{WP}}$, this formulation makes the computation less efficient as the input $x_{\text{COMP}}$ is no longer shared across different predicates. However, it offers a more tailored input signal and delivers above 99 F1 on both WSJ and Brown.

| Data | Model | WSJ VN | Brown VN |
|---|---|---|---|
| | | $x_{\text{WP}}$ | |
| Joint | Multitask | 83.14 | 86.26 |
| | Joint | **99.83** | **98.12** |
| Semi | Multitask | 93.08 | 85.71 |
| | Joint+CRF$_{\text{PB}}$ | 99.81 | 97.88 |
| | Marginal | 99.76 | 97.45 |
| | Marginal$_{\text{SEML}}$ | 99.68 | 97.71 |
| | | $x_{\text{COMP}}$ | |
| Joint | Joint | **99.85** | **99.02** |

Table 4: VN completion with gold PB labels. Results are averaged over models trained with 3 random seeds. $x_{\text{COMP}}$: Input construction for the completion mode.

## 7 Analysis

We report statistical metrics in Sec. 7.1. In Sec. 7.2, we analyze the use of constrained learning on the jointly labeled data.

## 7.1 Variance of SRL Models

A majority of F1s in Tab. 2 vary in a small range. Models trained on joint-only data show higher variance on the out-of-domain Brown test set. Among the semi-supervised models, the *Marginal* models exhibit high F1 variance on the Brown set while the *Marginal*$_{\text{SEML}}$ models work more stably.

## 7.2 Impact of Learning with SEMLINK Constraint on Joint Data

In Table 5, we use the form of constrained learning in Eq. 8 but apply it on the joint CRF loss over the jointly labeled training data. Note that the constraint term only affects the denominator part in Eq. 6. Overall, the effect of SEMLINK at training time seems small. On the WSJ test set, both VN and PB F1s are fairly close. The Brown test F1s have a drop, especially on VN, suggesting that constrained learning on the joint data is not needed.

| | Joint | WSJ | | Brown | |
|---|---|---|---|---|---|
| | SEMLINK | VN | PB | VB | PB |
| 1 | ✗ | 91.42 | 89.12 | 85.90 | 82.56 |
| 2 | ✓ | 91.42 | 89.21 | 85.41 | 82.46 |

Table 5: Ablation of SEMLINK constraint during training using the *Joint* CRF. ✓ indicates SEMLINK constraint is applied; and ✗ indicates not.

## 7.3 Reliance on Gold Predicate Labels

While we focused on experiments that use given gold predicate labels (i.e., VN class $\eta$ and PB sense $\sigma$), our models do not rely on them. Intuitively, there will be a performance degradation on argument labeling when predicate labels are not always accurate. To study this degradation curve, we experiment with randomly corrupted predicate classes/senses, to illustrate the performance dependency on predicate disambiguation. Specifically, we perturb the development split by randomly swapping a predicate's VN class (or PB sense) with another one from the dataset and observe how well our best model performs. This setup essentially simulates a real-world testing scenario where predicate labels on either VN/PB are imperfect. Results in Tab. 6 suggest that F1 degradation happens in a smooth way due to error propagation.

## 8 Discussion

The use of constraints in SRL has a long history that mostly focuses on the PropBank SRL task. Earlier

| $\eta$ corruption (%) | 0 | 5 | 10 | 20 | 30 |
|---|---|---|---|---|---|
| VN F1 | 91.23 | 89.45 | 83.64 | 82.90 | 79.02 |
| PB F1 | 88.82 | 88.23 | 87.32 | 85.56 | 84.66 |

| $\sigma$ corruption (%) | 0 | 5 | 10 | 20 | 30 |
|---|---|---|---|---|---|
| VN F1 | 91.23 | 90.35 | 89.55 | 87.18 | 84.94 |
| PB F1 | 88.82 | 88.17 | 87.21 | 84.92 | 83.11 |

Table 6: Performance curve w.r.t. predicate label corruption, measured using the *Marginal*$_{\text{SEML}}$ trained with random seed 1. $\eta$: VN class. $\sigma$: PB sense.

work investigated inference with constraints (e.g. Punyakanok et al., 2004; Surdeanu et al., 2007; Punyakanok et al., 2008). Other work developed constrained models for learning (e.g. Chang et al., 2012) ; or incorporated constraints with emerging neural models (e.g. Riedel and Meza-Ruiz, 2008; Fürstenau and Lapata, 2012; Täckström et al., 2015; FitzGerald et al., 2015; Li et al., 2020).

VerbNet SRL, on the other hand, is often studied as a comparison or a helper for PropBank SRL (Kuznetsov and Gurevych, 2020). Yi et al. (2007) showed that the mapping between VerbNet and PropBank can be used to disambiguate PropBank labels. It has been shown that model performance on VerbNet SRL is affected more by predicate features than PropBank SRL (Zapirain et al., 2008). In a sense, our observation that VerbNet F1 gains larger improvements from SEMLINK is also consistent with prior work. Beyond comparison work, Kazeminejad et al. (2021) explored the downstream impact of VerbNet SRL and showed promising uses in entity state tracking.

**Multitask Learning** The closest work to this paper is Gung and Palmer (2021). Instead of modeling argument labels and predicate classes via multitasking, we adopted a simpler design and focused on joint SRL argument labeling. This comes with two benefits: 1) a focused design that models joint labels; 2) an easy extension for using marginal CRF for partially labeled data. Other technical differences include a generally better transformer (i.e., RoBERTA) instead of BERT (Vaswani et al., 2017), simpler input construction, and our proposal of the completion mode.

**Marginal CRF** Greenberg et al. (2018) explored the use of marginal CRF on disjoint label sequences in the biomedical domain. Disjoint label sequences are concatenated into one, thus requiring dedicated decoding to reduce inconsistency w.r.t. various

structure patterns (e.g., aligned BIO pattern). In this paper, we took a step further by pairing label sequences to form a joint SRL task, allowing an easy interface for injecting decoding constraints.

**Broader Impact**    Recent advantages in large language models (LLM) have shown promising performance in semantic parsing tasks (e.g. Drozdov et al., 2022; Mekala et al., 2022; Yang et al., 2022). A well-established approach is via iterative prompting (e.g., Chain-of-Thought (Wei et al., 2022)) and potentially using in-domain examples for prompt construction. While such work bears many technical differences from this work, there are advantages that can potentially be shared. For instance, our direct use of a knowledge base (SEMLINK in our case) allowed for a guarantee of $0$ violations; and LLM-based work is less reliant on training data. Another scenario is when treating semantic structures as explicit intermediate products, such as for language generation. Our joint modeling allows for $\geqslant 99\%$ accuracy in converting PropBank arguments to VN arguments. When using such labels for prompted inference, it can make fewer errors.

**Conclusions**    In this work, we presented a model that learns from compatible label sequences for the SRL task. The proposal includes a joint CRF design, extension for learning from partially labeled data, and reasoning and learning with SEMLINK constraints. On the VerbNet and PropBank benchmark based on CoNLL05, our models achieved state-of-the-art performance with especially strong out-of-domain generalization. For the newly proposed task of completing VerbNet arguments given PropBank labels, our models are near perfect, achieving over 99 F1 scores.

## 9    Limitations

**Towards fully end-to-end parser.**    Our model architecture is on the track of end-to-end SRL parser, but it still assumes gold predicate positions and predicate attributes are given. A fully end-to-end parser can take sentences in raw text and output disjoint label sequences. While doing so can make computation less efficient (e.g., requiring substantially larger memory for training), it can bring users convenience.

**Involving document context.**    Gung and Palmer (2021) showed that using neighboring sentence prediction with transformer positively impacts parsing

F1. In contrast, we assumed sentences in the corpus are independent.

**Why does PropBank seem more difficult?**    We hypothesized the reason to be less granularity in argument labels and more ambiguous label assignments. As mentioned in Sec. 3, prior work benefited from using dedicated label text/definition as an auxiliary input. We only used such features at the predicate level, implicitly trading off potential gains on PB F1 for more efficient computation.

**Marginal model's capacity at handling constraints.**    In this paper, we focused on the SEMLINK constraint for compatible label sequences. There is a broad spectrum of SRL constraints in prior work (Punyakanok et al., 2008), some of them do not easily fit in the marginalization formulation, such as the *unique core role* constraint.

## Acknowledgement

We thank the reviewers for their helpful insights and comments. We gratefully acknowledge the support of NSF under grants #1801446 (SATC) and #1822877 (Cyberlearning), and of DARPA CwC (subcontracts from UIUC and SIFT), DARPA AIDA Award FA8750-18-2-0016 (RAMFIS), DARPA KAIROS FA8750-19-2-1004-A20-0047-S005 (RESIN, sub to RPI, PI: Heng Ji), as well as DTRA HDTRA1-16-1-0002/Project 1553695 (eTASC - Empirical Evidence for a Theoretical Approach to Semantic Components).

The views and conclusions contained herein are those of the authors and should not be interpreted as necessarily representing the official policies of any government agency.

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

## A Appendix

### A.1 Hyperparameters

As discussed in Sec. 6, we adopted a 2-stage fine-tuning for each of our models. In the first round, we fine-tune for 20 epochs with initial learning rate of $3 \times 10^{-5}$. In the second round, we fine-tune for another 5 epochs with fresh start on the optimizer. Across the two stages, we applied a dropout

layer with rate $0.5$ (which is preliminarily grid-searched from $[0.1, 0.2, 0.3, 0.4, 0.5, 0.6, 0.7, 0.8]$) before each of the linear layers in Sec. 3. The ROBERTA transformer uses the built-in default configurations which is implemented by Wolf et al. (2019).

The learning rate ($\lambda$) in the second round of fine-tuning varies by model. We preliminarily grid-searched from $[3 \times 10^{-6}, 1 \times 10^{-5}, 3 \times 10^{-5}, 1 \times 10^{-4}]$. Specifically, for each learning rate, we trained 3 random models and chose the one with the highest overall SRL F1 on the Dev set. We put their values in Table 7.

| Data | Model | $\lambda$ |
|---|---|---|
| Joint | Multitask | $1 \times 10^{-4}$ |
| | Joint | $1 \times 10^{-4}$ |
| Semi | Multitask | $1 \times 10^{-4}$ |
| | Joint+CRF$_{\text{PB}}$ | $1 \times 10^{-4}$ |
| | Marginal | $3 \times 10^{-5}$ |
| | Marginal$_{\text{SEML}}$ | $1 \times 10^{-5}$ |

Table 7: Learning rate in the second round of fine-tuning for each model in Table 2.