# OpenReview forum: "Learning Semantic Role Labeling from Compatible Label Sequences"
_EMNLP/2023/Conference — EMNLP 2023 Findings_

### Official Review · Reviewer_wA9b · 2023-07-28

**Soundness:** 4

**Excitement:**

3: Ambivalent: It has merits (e.g., it reports state-of-the-art results, the idea is nice), but there are key weaknesses (e.g., it describes incremental work), and it can significantly benefit from another round of revision. However, I won't object to accepting it if my co-reviewers champion it.

**Paper Topic And Main Contributions:**

This paper explores outputting multiple SRL label sequences with compatible labels, more specifically, jointly outputting PropBank (PB) and VerbNet (VN) labels. The authors propose a joint CRF model that outputs joint PB-VN labels, which is shown to be generally better than a multi-task baseline, especially on the out-of-domain Brown test set. Furthermore, semi-supervised learning with PB-only data and inferencing with SemLink constraints are explored, which could effectively bring benefits.

**Questions For The Authors:**

- A: Although the joint label space is pruned, it seems still pretty large, how would this affect training and testing efficiency? How slower would it be compared to the methods that predict separately?
- B: How effective is CRF in these settings? With strong neural models, maybe simple token-wise labelers would obtain good results (maybe adding BIO constraint enforcing at testing time)?
- C: The main experimenting dataset only covers 56% of the CoNLL05 predicates, are there any biases for these convertible predicates? For example, are they more frequent ones? Also, it would be better if more analysis can be provided based on predicate frequencies (my intuition is that probably low-frequent predicates would be the ones that benefit more from joint learning).


**Reasons To Accept:**

- I think the direction explored in this work is interesting and important for the future development of unifying semantic resources.
- The paper is well-written and easy to follow, and the experiments are well-conducted.


**Reasons To Reject:**

- The experiments only involve joint prediction of PB and VN, where there are abundant joint data and high-coverage mapping resources. The paper could be much stronger and more interesting if it could be extended to scenarios where there are less such joint resources (like PropBank/VerbNet & FrameNet).
- The performance gaps between different methods seem small. Though this is not quite surprising giving strong neural models, it would be more interesting to explore scenarios where the proposed methods could be more helpful (such as low-resource cases and less-frequent predicates).


**Reproducibility:**

4: Could mostly reproduce the results, but there may be some variation because of sample variance or minor variations in their interpretation of the protocol or method.

**Reviewer Confidence:**

3: Pretty sure, but there's a chance I missed something. Although I have a good feel for this area in general, I did not carefully check the paper's details, e.g., the math, experimental design, or novelty.

---

> ### Author Rebuttal · Authors · 2023-08-28
>
> Thank you for the detailed comments and insights on broadening the impact of our work. We will add the discussions and additional analysis to strengthen our work.
>
> > The experiments only involve joint prediction of PB and VN, where there are abundant joint data and high-coverage mapping resources. The paper could be much stronger and more interesting if it could be extended to scenarios where there are less such joint resources (like PropBank/VerbNet & FrameNet).
> - This is a good question. There are two challenges to joint predictions. One is how to max out model performances with existing joint data at hand. The other one is how to make it generalize to resources with less or no alignments. Our work focused on the first question.
>
> > The performance gaps between different methods seem small.
> - We genuinely believe a 3.4 (VN) improvement over the prior state-of-the-art is not a small one for parsing task. The momentum is from 83.0 to 86.4. Granted that our improvement on the PB axis is relatively smaller (0.8 over baseline and 0.2 over prior best).
>
> > Though this is not quite surprising giving strong neural models, it would be more interesting to explore scenarios where the proposed methods could be more helpful (such as low-resource cases and less-frequent predicates).
> - We believe this is indeed an interesting space to explore. It will possibly lead to similar observations in [1] that external structures bring a larger performance boost when there are very few annotated sentences.
> - We want to point out that our semi-supervised models are in a sense on such low-resource settings. There is a wider availability of PB-annotated sentences but much fewer VN-annotated ones. Our semi-supervised models benefit from those PB-only annotations to improve performances on the lower-resourced VN task.
> - We present analysis on performance over different predicate class frequency in our response to your Question C.
>
> [1]: Structured Tuning for Semantic Role Labeling. Li et. al. ACL 2020
>
> > Question A: Although the joint label space is pruned, it seems still pretty large, how would this affect training and testing efficiency? How slower would it be compared to the methods that predict separately?
> - The most noticeable part is the GPU memory consumption. In our multitask model, we have separate CRFs for VN and PB, each side has 100+ labels. In comparison, our joint label space is 685. With the same transformer backbone and the same CRF architecture, GPU memory is almost doubled for training. Accordingly, training time is also slowed by the relative larger label space. Our rough estimation is about 50% more training time when moving from multitask to joint model.
> - Inference speed is actually not noticeably slower, even though the complexity of Viterbi algorithm seems to suggest otherwise (O(Length * Label^2). This is likely due to the Viterbi decoding algorithm is already vectorized on the label dimension.
>
> > Question B: How effective is CRF in these settings? With strong neural models, maybe simple token-wise labelers would obtain good results (maybe adding BIO constraint enforcing at testing time)?
> - To the best of our knowledge, CRF and Viterbi is still common component for sequence labeling task, especially for BIO schemes. The BIO consistency is managed by the Viterbi decoding during testing, and is managed by CRF during training; since both of them are global optimizations sharing similar objectives.
> - Removing the CRF module, in our preliminary experience, always hurt the labeling performance by a large margin. The detailed ablation dates back to the introduction of LSTM+CRF[2] and such design is carried over onto BERT and RoBERTa encoders. The prior state-of-the-art (Gung and Palmer IWCS21) also followed this design.
>
> [2]: Bidirectional LSTM-CRF Models for Sequence Tagging. Huang et. al. 2015
>
> > Question C: The main experimenting dataset only covers 56% of the CoNLL05 predicates, are there any biases for these convertible predicates? For example, are they more frequent ones?
> - One potential “bias” could be the long-tailed distribution of predicates. This is indeed the case in our analysis below.
>
> > Question C: Also, it would be better if more analysis can be provided based on predicate frequencies (my intuition is that probably low-frequent predicates would be the ones that benefit more from joint learning).
> - We did an analysis on the validation split. Our finding (below) indicates that the average improvements tend to be "uniform" over predicate attribute frequencies. But both VN and PB improvements indeed tend to have higher variance at lower predicate frequency.
> - [vn_accuracy_diff](https://anonymous.4open.science/r/emnlp23_rebuttal-CCC6/vn_f1_diff.png) shows the VN accuracy gain of our best model over the joint multitask baseline. The x-axis indicates VN class distribution.The red line indicates the mean at each frequency.
> - [srl_accuracy_diff](https://anonymous.4open.science/r/emnlp23_rebuttal-CCC6/srl_f1_diff.png) shows the PB accuracy gain of our best model over the joint multitask baseline. The x-axis indicates PB sense distribution. The red line indicates the mean at each frequency.
> - Please note that the averaged improvements are relatively small on the validation set, thus margins are not always substantial in the above plots.

---

### Official Review · Reviewer_bCiH · 2023-08-04

**Soundness:** 4

**Excitement:**

3: Ambivalent: It has merits (e.g., it reports state-of-the-art results, the idea is nice), but there are key weaknesses (e.g., it describes incremental work), and it can significantly benefit from another round of revision. However, I won't object to accepting it if my co-reviewers champion it.

**Paper Topic And Main Contributions:**

This paper explores various joint traiing decoding strategies for Propbank and VerbNet based semantic role labeling.

**Reasons To Accept:**

The results are state of the art, the methods sensible and the discussion useful.

**Reasons To Reject:**

The existence of compatible verbnet and PropBank labels is an accident of the history of the field There are not many situations parallel to this, so the result is probably of interest only to a small subset of ACL researchers. The authors make little attempt to draw general lessons from their work.

**Reproducibility:**

5: Could easily reproduce the results.

**Reviewer Confidence:**

4: Quite sure. I tried to check the important points carefully. It's unlikely, though conceivable, that I missed something that should affect my ratings.

---

> ### Author Rebuttal · Authors · 2023-08-28
>
> Thank you for appreciating our contributions. We will add the discussion to strengthen our work.
>
> > The existence of compatible verbnet and PropBank labels is an accident of the history of the field There are not many situations parallel to this, so the result is probably of interest only to a small subset of ACL researchers. The authors make little attempt to draw general lessons from their work.
> - The fuzzy alignment between PropBank and VerbNet is only one of the many bonus features offered by SemLink. SemLink also includes mappings between VerbNet and FrameNet, as well as direct mappings between PropBank and FrameNet.  Future work will explore the adaptation of this approach to FrameNet parsing, to determine how well it generalizes.  It could also be applied to Abstract Meaning Representations, which rely on the same PropBank frame files as SRL, but with different assumptions about applicability.
> - Here are some insights our work could offer (in addition to those in the Broader Impact section):
>   - The mutual benefit of VN and PB promotes a unified design of semantic structure/resource.
>   - Our use of consistency between label sequences to improve tagging accuracy conceptually connects to the idea of using consistency to improve LLM's prompted inference, although the modeling strategies are very different. We believe our work offers a positive signal to upcoming SRL models that are based on prompted inference of LLM: labeling consistency is of great potential.

---

### Official Review · Reviewer_Evjp · 2023-08-09

**Typos Grammar Style And Presentation Improvements:** There is a missing period at line 383.
**Soundness:** 4

**Excitement:**

3: Ambivalent: It has merits (e.g., it reports state-of-the-art results, the idea is nice), but there are key weaknesses (e.g., it describes incremental work), and it can significantly benefit from another round of revision. However, I won't object to accepting it if my co-reviewers champion it.

**Paper Topic And Main Contributions:**

This paper presents a joint model that simultaneously addresses two distinct SRL tasks with different formats, namely VerbNet SRL and PropBank SRL.

By incorporating SEMLINK (Stowe et al. , 2021) during decoding, the authors successfully prevent the generation of conflicting argument labels.

The experimental results indicate that the method proposed by the authors outperforms previous multitasking learning approaches.
And when provided with the gold standard PropBank SRL labels, this method achieves an F1 score of 99 for predicting VerbNet SRL labels.

**Questions For The Authors:**

Question A：When constructing "x_wp,"  did you predict the predicates and senses first or did you directly use the correct answers? If the predicted answers were used, does that imply that both "x" and "x_wp" underwent separate encoding, with the encoding result of "x" being utilized for predicting predicates?

Question B: In line 406, you mentioned that correct VerbNet classes and PropBank senses were used during constrained decoding. Have you attempted using predicted answers?

**Reasons To Accept:**

Experiments conducted within this study are comprehensive, accompanied by a thorough analysis.

The proposed approach can be employed to facilitate mutual transformation between SRL data annotated under different standards, which holds significance for the automated construction of annotated datasets.

**Reasons To Reject:**

The writing of this paper requires a little improvement, as certain sections are difficult to comprehend. The foundational aspect of SEMLINK is inadequately introduced. For instance, in section 5, readers less familiar with SEMLINK might struggle to grasp concepts like "SEML(u)" and the associated inferences. I believe the authors could enhance understanding by incorporating more illustrative examples.

Additionally, as mentioned by the authors in the Limitation section, the model in this study relies on gold predicate positions and predicate attributes. This limitation diminishes the practicality of the model in real-world scenarios.

**Reproducibility:**

4: Could mostly reproduce the results, but there may be some variation because of sample variance or minor variations in their interpretation of the protocol or method.

**Reviewer Confidence:**

4: Quite sure. I tried to check the important points carefully. It's unlikely, though conceivable, that I missed something that should affect my ratings.

---

> ### Author Rebuttal · Authors · 2023-08-28
>
> Thank you for the detailed review and suggestions. We have added more illustration and analysis. We will such these evidences to strengthen our work.
>
> >The writing of this paper requires a little improvement, as certain sections are difficult to comprehend. The foundational aspect of SEMLINK is inadequately introduced. For instance, in section 5, readers less familiar with SEMLINK might struggle to grasp concepts like "SEML(u)" and the associated inferences. I believe the authors could enhance understanding by incorporating more illustrative examples.
> - We will add examples to motivate and better illustrate the structures of SemLink. One such example can be the predicate “value”:
> | PB sense | VN class | Admissible alignments |
> |-----|-----|-----|
> | 01 | price-54.4 | Arg0-Agent; Arg1-Theme; Arg2-Value |
> | 02 | price-54.4 | Arg0-Agent; Arg1-Theme; Arg2-Value |
> | 02 | admire-31.2 | Arg0-Experiencer; Arg1-Stimulus |
>
> - With a given predicate, each PB sense can be associated with multiple VN classes; and each VN class can also be associated with multiple PB senses. The PB sense and VN class jointly determine the set of admissible core arguments. Our models are therefore built to use such structure for improved parsing.
>
> > Additionally, as mentioned by the authors in the Limitation section, the model in this study relies on gold predicate positions and predicate attributes. This limitation diminishes the practicality of the model in real-world scenarios.
> - We want to clarify that our work focuses on experiments that use gold predicate attributes, but **does not rely on** them. Indeed, our experiments focused on scenario where such gold predicate features are given.
> - We chose such focus since prior works (e.g., Gung and Palmer IWCS21) have shown that predicate sense disambiguation tends to give high accuracy (e.g., >97% on WSJ and >91% on Brown in their paper).
> - Below, we show how our best model performs when predicate attributes are noisy. We experimented with randomly corrupted predicate attributes, to illustrate the performance dependency on predicate sense disambiguation. Specifically, we perturb the validation dataset by randomly swapping a predicate’s VN class (or PB sense) with another one from the dataset and observe how well our trained model performs on the validation set.
>   - VN class corruption
> | VN class corruption percent | 0% | 5% | 10% | 20% | 30% |
> |-----|-----|-----|-----|-----|-----|
> | VN F1 | 91.23 | 89.45 | 87.64 | 82.90 | 79.02 |
> | PB F1 | 88.82 | 88.23 | 87.32 | 85.56 | 84.66 |
>   - PB sense corruption
> | PB sense corruption percent | 0% | 5% | 10% | 20% | 30% |
> |-----|-----|-----|-----|-----|-----|
> | VN F1 | 91.23 | 90.35 | 89.55 | 87.18 | 84.94 |
> | PB F1 | 88.82 | 88.17 | 87.21 | 84.92 | 83.11 |
>   - This setup essentially simulates a real-world application scenario where predicate attributes are 100%, 95%, 90%, 80%, or 70% accurate accordingly. This study suggests that our model can use non-perfect predicate attributes. The resulting performance degradation happened in a smooth way and is expected due to error propagation.
>
> > Question A: When constructing "x_wp," did you predict the predicates and senses first or did you directly use the correct answers?
> - When constructing the wp, we used the gold predicate attributes.
>
> > Question A: If the predicted answers were used, does that imply that both "x" and "x_wp" underwent separate encoding, with the encoding result of "x" being utilized for predicting predicates?
> - One option is to predict predicate senses from “x” and then use such senses to construct “x_wp”. In this case, yes, they undergo separate encodings.
> - Another option is to build an end-to-end differentiable pipeline that handles predicate senses as well as the SemLink structures in a “soft” way. This option would require substantial model changes.
>
> > Question B: In line 406, you mentioned that correct VerbNet classes and PropBank senses were used during constrained decoding. Have you attempted using predicted answers?
> - We experiment with predicting using corrupted predicate attributes (in the above table). The randomly corrupted predicate attributes affect the construction of x_wp and the associated argument alignment constraints. We expect to see both VN and PB performances will be dragged by such corruption in a smooth manner. This is indeed the case as shown in the table.

---

### Meta-Review · Area_Chair_wh9q · 2023-09-19

**Recommendation:** 4

**Metareview:**

This paper proposes joint training and decoding strategies for Propbank and VerbNet-based semantic role labeling through the incorporation of SemLink. Experiments show SRL predictions to be on par or better than previous approaches.

Soundness:
All reviewers agree on their "soundness" score of 4, i.e. that the presented experiments are comprehensive, the discussion is informative, and the presented results support the conclusions of the paper. This points to the potential of using two different SRL formalisms to support each other in decoding.

However, it is also pointed out that some improvements are rather marginal. While Tables 1 and 2 present experiments averaged over three runs, they do not present a standard deviation score. Some of the bolded results in these tables are most certainly not statistically significant (for instance WSJ Joint decoding in Table 1 for PB increasing from 89.10 to 89.12). Still, as authors point out in their response, depending on the dataset and SRL type, some improvements are larger.

Another challenge to soundness is that only a single dataset is considered that only covers 56% of the CoNLL05 predicates. The authors discuss in their response how this might bias the results.

Excitement:
All reviewers agree on an "excitement" score of 3. The main challenge to excitement is the narrow focus of the paper on two particular SRL formalisms linked by SemLink, the focus on a single dataset with derived VN predicates, and the moderate improvements in decoding scores. The authors argue in their response that their results in principle may transfer to other SRL resources like FrameNet, but no supporting experiments are presented in the paper.

Clarity:
Reviewer agree that the paper is interesting to read, but could be improved for readers who are not familiar with details on the formalisms (more examples, better explanation of SemLink, concepts like "SEML(u)", etc.) The reviewers provide some clarification in their response. Should the paper be accepted, I would recommend clarifying these aspects (and perhaps even working with a Figure to better illustrate the overlap between the two formalisms.)

---

### Decision · Program_Chairs · 2023-10-07

**Decision:**

Accept-Findings

**Comment:**

This paper proposes joint training and decoding strategies for Propbank and VerbNet-based semantic role labeling through the incorporation of SemLink. Experiments show SRL predictions to be on par or better than previous approaches.

Soundness:
All reviewers agree on their "soundness" score of 4, i.e. that the presented experiments are comprehensive, the discussion is informative, and the presented results support the conclusions of the paper. This points to the potential of using two different SRL formalisms to support each other in decoding.

However, it is also pointed out that some improvements are rather marginal. While Tables 1 and 2 present experiments averaged over three runs, they do not present a standard deviation score. Some of the bolded results in these tables are most certainly not statistically significant (for instance WSJ Joint decoding in Table 1 for PB increasing from 89.10 to 89.12). Still, as authors point out in their response, depending on the dataset and SRL type, some improvements are larger.

Another challenge to soundness is that only a single dataset is considered that only covers 56% of the CoNLL05 predicates. The authors discuss in their response how this might bias the results.

Excitement:
All reviewers agree on an "excitement" score of 3. The main challenge to excitement is the narrow focus of the paper on two particular SRL formalisms linked by SemLink, the focus on a single dataset with derived VN predicates, and the moderate improvements in decoding scores. The authors argue in their response that their results in principle may transfer to other SRL resources like FrameNet, but no supporting experiments are presented in the paper.

Clarity:
Reviewer agree that the paper is interesting to read, but could be improved for readers who are not familiar with details on the formalisms (more examples, better explanation of SemLink, concepts like "SEML(u)", etc.) The reviewers provide some clarification in their response. Should the paper be accepted, I would recommend clarifying these aspects (and perhaps even working with a Figure to better illustrate the overlap between the two formalisms.)